# SIRT3 Acts as a Positive Autophagy Regulator to Promote Lipid Mobilization in Adipocytes via Activating AMPK

**DOI:** 10.3390/ijms21020372

**Published:** 2020-01-07

**Authors:** Tian Zhang, Jingxin Liu, Qiang Tong, Ligen Lin

**Affiliations:** 1State Key Laboratory of Quality Research in Chinese Medicine, Institute of Chinese Medical Sciences, University of Macau, Avenida da Universidade, Taipa, Macau 999078, China; yb67520@um.edu.mo (T.Z.); liujingxin@sztu.edu.cn (J.L.); 2Children’s Nutrition Research Center, Baylor College of Medicine, Houston, TX 77030, USA; qtong@bcm.edu; 3Zhuhai UM Science & Technology Research Institute, Zhuhai 519031, Guangdong, China

**Keywords:** sirtuin 3, lipid metabolism, macroautophagy, chaperone-mediated autophagy, AMPK

## Abstract

Obesity is increasing at an alarming rate worldwide, which is characterized by the excessive accumulation of triglycerides in adipocytes. Emerging evidence has demonstrated that macroautophagy and chaperone-mediated autophagy (CMA) regulate lipid mobilization and play a key role in energy balance. Sirtuin 3 (SIRT3) is an NAD^+^-dependent deacetylase, which is important in regulating macroautophagy and lipid metabolism. It is still unknown whether SIRT3 modulates macroautophagy and CMA in adipocytes. The current study found that macroautophagy was dynamically regulated during 3T3-L1 adipocyte differentiation, which coincided with SIRT3 expression. In mature adipocytes, overexpression of SIRT3 activated macroautophagy, mainly on lipid droplets (LDs), through activating the AMP-activated protein kinase (AMPK)-unc-51-like kinase 1 (ULK1) pathway, which in turn resulting in smaller LD size and reduced lipid accumulation. Moreover, SIRT3 overexpression induced the formation of perilipin-1 (PLN1)-heat shock cognate 71 kDa protein (HSC70)-lysosome-associated membrane protein 2 (LAMP2) complex, to activate CMA and cause the instability of LDs in adipocytes. In summary, we found SIRT3 is a positive regulator of macroautophagy and CMA in adipocytes, which might be a promising therapeutic target for treatment of obesity and its related metabolic dysfunction.

## 1. Introduction

The global prevalence of obesity has increased substantially over the past decades [1], which is positively relevant to cardiovascular diseases [2], type 2 diabetes [3], and certain types of cancer [4]. Adipocyte dysfunction is a key character during the onset and development of obesity. The storage and mobilization of lipids in adipocytes are essential to maintain energy balance. Under positive energy conditions, the excessive energy is converted into triglycerides (TGs) and deposited in lipid droplets (LDs) in adipocytes, resulting in the expansion of white adipose tissue (WAT) [5,6]. Under negative energy conditions, LD-sequestered TGs are hydrolyzed by cytosolic lipases to release free fatty acids (FFAs), which subsequently undergo β-oxidation to supply energy [7]. Alternatively, LDs are engulfed into autolysosomes to release FFAs, a process termed lipophagy [8]. Inhibition of macroautophagy increases lipid storage in hepatocytes [8,9]. Ablation of a macroautophagy-related gene decreases fat mass and protects mice from diet-induced obesity [10]. By contrast, a moderate activation of macroautophagy leads to decrease the body weight of mice [11]. These studies demonstrate the complicated roles of macroautophagy in regulating lipid storage and mobilization. Moreover, macroautophagy regulates cytoplasmic remodeling and mitochondrial turnover in the process of adipogenesis. Suppression of macroautophagy blocks adipogenesis, and reduces the formation of large LDs in adipocytes [12,13]. Targeted deletion of *Atg5* or *Atg7* in adipocytes exhibits anti-obesity and anti-diabetic effects [12,14]. 3-Methyladenine (3-MA), a macroautophagic inhibitor, stimulates lipolysis in adipocytes [15]. To date, the involvement of macroautophagy in adipogenesis and lipid turnover is still controversial.

Emerging evidence indicates that chaperone-mediated autophagy (CMA) participates in lipid metabolism [16]. In the process of CMA, the substrates with a pentapeptide motif are recognized by the heat shock cognate protein of 70 kDa (HSC70), which are subsequently delivered to the surface of lysosomes to bind to lysosome-associated membrane protein 2 (LAMP2) [17]. CMA selectively degrades the perilipin (PLIN) family proteins, the key coating proteins on LD surface [18,19]. PLIN1 is mainly expressed in adipocytes, whereas PLIN2 and PLIN3 are ubiquitously expressed [20]. Intriguingly, AMP-activated protein kinase (AMPK)-dependent phosphorylation of PLINs activates CMA, which is essential for the initiation of lipid mobilization via either cytosolic lipases or macroautophagy [21].

Sirtuins (SIRTs) are a family of NAD^+^–dependent deacetylases with a wide variety of physiological and pathological functions [22]. SIRT3 is mainly localized in mitochondria, which modulates the deacetylation of many substances [23]. *Sirt3* deficiency inhibits macroautophagy and disrupts mitochondrial homeostasis in cardiac aging mice [24,25]. Overexpression of *Sirt3* induces macroautophagy and CMA to protect against lipotoxicity in hepatocytes [26]. *Sirt3* overexpression activates macroautophagy to prevent mitochondrial injury and cardiomyocyte apoptosis [27]. By contrast, a recent study showed *Sirt3* silencing enhances macroautophagy flux in hepatocytes [28]. Until now, the role of SIRT3-mediated macroautophagy in adipocytes has been unrevealed.

To test the hypothesis that SIRT3 promotes lipid mobilization in adipocytes by enhancing macroautophagy and CMA, we compared the dynamic changes of macroautophagy and SIRT3 expression during adipocyte differentiation, and evaluated the effect of *Sirt3* overexpression on lipid mobilization in adipocytes and its underlying mechanisms.

## 2. Results

### 2.1. Macroautophagy and SIRT3 Expression are Dynamically Regulated During Adipocyte Differentiation

3T3-L1 fibroblasts are well-characterized preadipocyte cells, which can be induced to differentiate into typical white adipocytes [29]. The differentiation procedure of 3T3-L1 cells was shown in Figure 1a. To determine the change pattern of macroautophagy during differentiation, the 3T3-L1 cells lysates were harvested at different time points. As shown in Figure 1b, macroautophagy was induced at the initial stage of differentiation and peaked at two days post hormonal stimulation, indicated by the increased LC3-II/LC3-I ratio and Beclin1 expression, and the decreased p62 expression. Thereafter, macroautophagy was suppressed and maintained at a low level, indicated by the decreased LC3-II/LC3-I ratio and Beclin1 expression, and the increased p62 expression. These results indicated that macroautophagy was dynamically regulated and highly involved in adipocyte differentiation. Interestingly, the SIRT3 expression was relatively high at the initial stage of adipocyte differentiation and declined from four days post hormonal stimulation, which coincided with the change pattern of macroautophagy (Figure 1b).

To directly evaluate the role of SIRT3 in regulating macroautophagy in adipocytes, we generated a 3T3-L1 cell line expressing the full-length *Sirt3* cDNA (SIRT3OE). SIRT3OE cells expressed around 2.5-fold more SIRT3 protein than that expressed in the cells transfected with pcDNA3.1 plasmid (vector); and the SIRT3 protein was mainly expressed in cytosol, but not the nucleus (Figure 1c). As expected, the conversion of LC3-I to LC3-II and Beclin1 expression were increased, and p62 expression was decreased in SIRT3OE cells (Figure 1d), suggesting SIRT3 overexpression activated macroautophagy in adipocytes. To further confirm this, SIRT3OE and vector cells were infected with an mRFP-GFP-LC3 tandem construct, and the colocalization of mRFP-LC3 and GFP-LC3 puncta was detected. When autophagosomes merge with lysosomes, GFP can be quenched easily in acidic environments and loses its fluorescence in autolysosomes. While mRFP red fluorescence is stable under acidic conditions and can mark and track LC3 during autophagy process. More red-only puncta were observed in SIRT3 overexpressed cells, which was similar to the adipocytes starved for 16 h (Figure 1e), indicating SIRT3 overexpression enhanced autophagic flux in adipocytes without disrupting the lysosomal function and/or autophagosome-lysosome fusion. In the LC3 turnover assay, the difference in LC3-II level in the presence or absence of 3-MA was greater in SIRT3OE cells (Figure 1f), which further indicated the macroautophagic flux was enhanced in SIRT3OE adipocytes. Furthermore, a p62 turnover assay treated with or without rapamycin resulted in an observed reduction of Beclin1 and LC3-II protein level in the *Sirt3* silenced adipocytes (Appendix A). Taken together, the above data suggest that SIRT3 overexpression enhances macroautophagic flux in adipocytes.

### 2.2. Overexpression of SIRT3 Decreases Lipid Accumulation in Adipocytes through Enhancing Lipophagy

Oil Red O staining results revealed reduced lipid accumulation in SIRT3OE adipocytes, compared with the vector control cells (Figure 2a). In mature adipocytes, 24 h 3-MA treatment obviously increased lipid content in both vector and SIRT3OE cells, and partially abolished SIRT3 overexpression-induced reduction of lipid accumulation in adipocytes (Figure 2a). On the contrary, 16 h starvation resulted in significant reduction of lipid content in both vector and SIRT3OE cells (Figure 2a). Next, SIRT3 overexpression in adipocytes significantly decreased TG and total cholesterol (T-CHO) levels, which were almost reversed by 3-MA treatment (Figure 2b). 3-MA is considered as a non-specific macroautophagy inhibitor. Thus, siRNA-mediated *Atg5* knockdown was performed on both vector and SIRT3OE cells, with the knockdown efficiency around 75% (Figure 2c and Appendix A). Knockdown of *Atg5* almost reversed SIRT3-induced macroautophagy (Figure 2c) and reduction of cellular TG and lipid contents (Figure 2d).

To confirm whether SIRT3-activated macroautophagy mainly occurred in LDs, Nile red staining was performed in SIRT3 and GFP-LC3 co-transfected 3T3-L1 cells. The results demonstrated SIRT3OE cells contained more small LDs and fewer big LDs, compared with the vector cells (Figure 2e). After 24 h 3-MA treatment, more big LDs were observed in mature adipocytes, for both vector and SIRT3OE cells, when compared with the corresponding vehicle control cells (Figure 2e). Interestingly, LC3 was significantly induced and co-localized with LDs in SIRT3OE cells, which was partially reversed by 3-MA treatment (Figure 2e). Next, the LD fraction was isolated from mature adipocytes, and an enrichment of LC3-II was observed in isolated LDs, but not in homogenates in SIRT3OE cells (Figure 2f). To further confirm that SIRT3 regulates the delivery of LD content to lysosomes, the LAMP2 level was evaluated in vector and SIRT3OE cell using immunostaining, with or without the lysosomal inhibitor, NL. In SIRT3OE cells, the LAMP2 expression was obviously increased, and the LAMP2 was highly co-localized with LDs (Figure 2g). When treated with NL, more big LDs were observed in both vector and SIRT3OE cells, and the induction of LAMP2 expression in SIRT3OE cells was completely reversed (Figure 2g). As expected, SIRT3 overexpression-induced reduction of TG and T-CHO levels were almost reversed by NL treatment (Figure 2h). Thus, SIRT3 overexpression decreased lipid accumulation and reduced LD size in adipocytes through enhancing lipophagy.

### 2.3. SIRT3 Induces Macroautophagy Through Activating AMPK in Adipocytes

The AMPK-FOXO3a axis plays a central role in energy homeostasis [30], and coordinates metabolism with macroautophagy [31]. Herein, the phosphorylation levels of AMPK and the downstream signal FOXO3a were significantly up-regulated by SIRT3 overexpression, which was blocked by treatment of Compound C (CC), an AMPK inhibitor (Figure 3a). FOXO3a activity is negatively associated with the acetylation level. The ratio of Ac-FOXO3a to FOXO3a was greatly decreased in SIRT3 overexpressed cells (Figure 3b). Next, the LC3 and Nile red staining was performed in SIRT3OE and vector cells, treated with or without CC. After 24 h CC treatment, more big LDs were observed in mature adipocytes, for both vector and SIRT3OE cells, when compared with the corresponding vehicle control cells (Figure 3c). In SIRT3OE cells, the LC3 expression was obviously increased and highly co-localized with LDs, which were completely reversed by CC (Figure 3c). The contents of TGs and T-CHO were significantly decreased in SIRT3 overexpressed adipocyte, which was almost salvaged by CC treatment (Figure 3d). These results indicated that SIRT3-mediated macroautophagy attenuates lipid accumulation in adipocytes via AMPK-FOXO3a pathway.

AMPK complex consists a catalytic subunit (α subunit) and two regulatory subunits (β and γ subunits). AMPKα1, AMPKβ1, and AMPKγ1 are ubiquitously expressed, while, AMPKα2 and AMPKβ2 are expressed in many other tissues at relatively low levels. SiRNA-mediated knockdown of *AMPKα1*, *AMPKα2*, *AMPKβ1* and *AMPKβ2* were performed on vector and SIRT3OE cells, with the knockdown efficiencies ranging from 78% to 85% (Figure 4a and Appendix A‒e). Knocking down *AMPKα1*, but not *AMPKα2*, totally abolished SIRT3-mediating reduction of lipid and TG contents (Figure 4b,c). Moreover, knocking down *AMPKβ1* but not *AMPKβ2*, partially reversed the effect of SIRT3 (Figure 4d,e). Intriguingly, *AMPKα1* silencing totally reversed SIRT3-induced activation of macroautophagy (Figure 4f). The phosphorylation levels of AMPK and FOXO3a were significantly up-regulated by SIRT3 overexpression, which was blocked by knocking down *AMPKα1* (Figure 4f). Consistently, knocking down *AMPKα1* completely reversed SIRT3OE-induced increase of LC3 expression and its co-localization with LDs (Figure 4g).

The unc-51-like kinase 1 (ULK1) plays a key role in macroautophagy induction, and AMPK directly phosphorylates and activates ULK1 to enhance macroautophagy. Herein, overexpressed SIRT3 upregulated the phosphorylated ULK1 level in adipocytes (Figure 5a). To further verify that SIRT3 promoted macroautophagy via the AMPK-ULK1 axis, MRT68921, a potent ULK1/2 inhibitor, was recruited. As shown in Figure 5a, MRT68921 treatment decreased the LC3-II/LC3-I ratio and Beclin1 level, and increased p62 level, to reverse SIRT3-induced activation of macroautophagy. Moreover, MRT68921 treatment also greatly increased lipid and TG accumulation in SIRT3OE adipocyte (Figure 5b,c). Furthermore, the immunostaining results indicated that treatment of MRT68921 almost reversed SIRT3-induced reduction of lipid accumulation and increase of LC3 expression on LDs (Figure 5d). These results indicated that SIRT3-mediated macroautophagy attenuates lipid accumulation through AMPK-ULK1 pathway.

### 2.4. SIRT3 Activates CMA Through Phosphorylating AMPK

CMA is critical for the initiation of lipid mobilization via either lipolysis or lipophagy. A previous study showed that the AMPK-dependent phosphorylation of PLIN2 is critical for its interaction with HSC70 [21]. Currently, the phosphorylated PLIN1 was detected by phos-tag SDS-PAGE. The results showed an increase of PLIN1 phosphorylation in SIRT3OE adipocytes, compared with that in vector cells, which was almost blocked by CC treatment (Figure 6a). When immuno-precipitated PLIN1, a lower level of bound HSC70 and a higher level of bound LAMP2 were observed in SIRT3OE cells (Figure 6b). When treated with CC, an increased level of bound HSC70 and lower level of bound LAMP2 were observed in the vector cells, and CC treatment partially reversed the changes in SIRT3OE cells (Figure 6b). In isolated LDs, the decreased levels of HSC70 and PLIN1 were observed in SIRT3OE cells, which were partially reversed by CC treatment (Figure 6c).

To confirm the role of CMA in SIRT3-mediating reduction of lipid accumulation, LAMP-2A was silenced on vector and SIRT3OE cells by using siRNA targeting *LAMP-2A*, with the efficiency of 83% (Figure 6d). Interestingly, *LAMP-2A* silencing reversed SIRT3-induced reduction of lipid and TG contents in adipocytes (Figure 6e,f). To further elucidate the role of macroautophagy and CMA in SIRT3-mediated reduction of lipid accumulation, simultaneous knockdown of *Atg5* and *LAMP-2A* was performed on vector and SIRT3OE cells. As expected, simultaneous silencing of *Atg5* and *LAMP-2A* resulted in more severe lipid accumulation in adipocytes, compared with either *Atg5* knockdown or *LAMP-2A* knockdown cells, and almost reversed the SIRT3’s effect on reduction of lipid accumulation in adipocytes, as assessed by both intracellular TG content and Nile red staining (Figure 6g−i). The above results demonstrated that SIRT3 promotes CMA facilitation of LD degradation through activating AMPK.

## 3. Discussion

Emerging evidence has revealed a close connection between macroautophagy and lipid metabolism. Cells and organisms modulate FFAs supply to adapt nutrient and metabolic demand through regulating macroautophagy [32]. Impaired macroautophagy leads to excessive lipid accumulation in the liver to cause hepatic steatosis [33], and alters fatty acid metabolism to reduce mass of WAT [14]. A moderately overexpressed Atg5 is sufficient to activate macroautophagy and decrease the body weight of mice, emphasizing a possible role for macroautophagy in lipid disposal [11]. The present study showed that macroautophagy is involved in adipocyte differentiation and activation of macroautophagy decreases lipid accumulation in adipocytes. On the other hand, intracellular lipids themselves also regulate macroautophagy. HFD feeding suppresses autophagy in mouse liver [8] and excess free cholesterol induces the activation of autophagy in smooth muscle cells as a cellular defense mechanism [34]. Our data showed that HFD feeding resulted in inactivation of macroautophagy, indicated by the decreased LC3II/LC3I ratio and Beclin1 expression, and the increased p62 expression, and suppression of CMA, indicated by the decreased LAMP2A expression, and the increased PLIN1 expression, in WAT from mice (Appendix A). These results indicated that macroautophagy and CMA promote lipid mobilization and reduce lipid accumulation in adipocytes, and the reduced lipid content, in turn, might further activate macroautophagic flux, to form a positive regulatory loop.

In contrast, a recent report showed that saturated fatty acids (SFAs)-induced activation of autophagy contributes to lipotoxicity in hepatocytes [28]. A previous study showed that SFAs inhibit fusion between autophagosomes and lysosomes and ameliorate degradation of autophagosomes, resulting in the blocked macroautophagic flux [35]. SFAs induce lipotoxicity, but not the adipogenesis condition, which might explain the controversial observations in the above study.

Macroautophagy does not only modulate lipid mobilization, but also regulates the adipogenic process. Numerous studies have demonstrated that macroautophagy is required for normal adipocyte differentiation. Knockout of either *Atg5* or *Atg7* results in dramatic suppression of adipocyte differentiation and a decrease of TG accumulation in preadipocytes [12,13,14]. In the current study, we found that adipocyte differentiation correlates with enhanced macroautophagic flux in the initial stage, and with suppressed macroautophagy in the terminal stage. The mitochondrial network morphology differs in pre- and differentiated adipocytes. Macroautophagy is required for the differentiation-specific remodeling of the mitochondrial network in 3T3-L1 cells, especially in the initial stage of differentiation (first two days) [36]. Suppression of macroautophagy blocks adipogenesis and the formation of big LDs in adipocytes [12], while a moderate activation of macroautophagy leads to decreased adipose tissue mass [11]. Thus, inhibition of macroautophagy suppresses the differentiation of preadipocytes into adipocytes at the initial stage; while, activation of macroautophagy increases lipid mobilization and reduces the accumulation of LDs in adipocytes at the terminal stage. As expected, the expression levels of adipogenic marker, peroxisome proliferator-activated receptor γ (PPARγ), and glucose transporter 4 (GLUT4), were increased in SIRT3OE cells (Appendix A). In the vector cells, 3-MA treatment at the initial stage of differentiation (day 0 to 2) obviously reduced lipid accumulation; while, 3-MA treatment at the terminal stage of differentiation (day 8−10), enhanced lipid accumulation (Appendix A). Moreover, 3-MA treatment blocked the effect of SIRT3 overexpression on lipid accumulation at either the initial or terminal stage of differentiation (Appendix A). These results indicated SIRT3 attenuates lipid accumulation in adipocytes, which is more than likely through enhancing lipid mobilization at the terminal stage of differentiation.

Although lipolysis and lipophagy share similar function, the interaction of these two processes is still unclear [8,37]. A recent study indicated that lipolysis and lipophagy are not completely independent pathways but actually regulated by each other. Activation of macroautophagy not only shifts lipids to the lysosome for degradation by acid lipases, but enhances lipolysis by neutral lipases [38]. SIRT3 overexpression did not induce lipolysis in the basic state, but stimulated lipolysis under isoproterenol-treated conditions (Appendix A). Further study showed that treatment of the lipolysis inhibitor, DEUP, partially blocked SIRT3-induced reduction of TG content in adipocytes (Appendix A), which suggested SIRT3 reduces lipid accumulation through both lipophagy and lipolysis.

SIRT3 is mainly localized in mitochondria. Mitochondria interact with LDs to increase the supply of FAs for mitochondrial β-oxidation, which requires the activation of AMPK [39]. Our data indicated that SIRT3 overexpression enhanced maximum oxygen consumption rate (OCR, Appendix A) and the expression of uncoupling protein 1 (UCP1, Appendix A) in adipoctyes. Under normal status, big LDs accounted for the majority and mitochondria were only distributed in a discrete region of the LDs in vector cells. In SIRT3OE adipocytes, multilocular LDs were observed, and parts of them became fully trapped in mitochondria (Appendix A). In contrast, under glucose starvation status, the LDs in SIRT3 overexpressed adipocytes were more susceptible to interaction with mitochondria on detyrosinated microtubules, compared with those of the vector cells (Appendix A). On the basis of morphometric analysis and changes in lipid profile, SIRT3 might facilitate the lipid mobilization by reorganizing the network of microtubules to facilitate consumption of lipids.

AMPK is a kinase that responds to mitochondrial energetics and regulates macroautophagic flux [28,40,41]. Phosphorylation of AMPK directly activates proteins involved in the initiation and development of macroautophagy, such as ULK1 and Beclin1 [42]. Furthermore, AMPK phosphorylates the transcription factor FOXO3, leading to increased expression of autophagy-related proteins. Knockdown of SIRT3 resulted in decreased macroautophagy through phosphorylation of AMPK in stressed peritoneal macrophages [40]. SIRT3 deficiency inhibited macroautophagy through AMPK in skeletal muscles of *SIRT3^−/−^* mice and C2C12 myotubes [43]. The current data showed that gain of SIRT3 expression increased the phosphorylation of AMPK and FOXO3, and its downstream signal ULK1, which, in turn, induced macroautophagy in adipocytes.

CMA promotes the degradation of PLIN, resulting in the exposure of LD core to neutral lipases [21]. During starvation, CMA is activated, which leads to the gradual degradation of PLIN2 and 3 in mouse fibroblasts. Subsequently, lipases and autophagic components are recruited to the surface of LDs, thereby increasing the rate of lipid mobilization [18]. In contrast, inhibition of CMA prevents LD degradation and lipolysis [21]. Moreover, AMPK phosphorylates PLINs to trigger degradation of PLIN1, which is the first step towards initiating cytosolic lipases- or macroautophagy-mediated lipolysis. The current data demonstrated SIRT3 activates CMA to induce LDs breakdown through AMPK-mediated PLIN1 phosphorylation.

## 4. Materials and Methods

### 4.1. Materials

Dulbecco’s Modified Eagle Medium (DMEM), phosphate-buffered saline (PBS), fetal bovine serum (FBS), 0.25% (w/v) trypsin-EDTA, and penicillin-streptomycin (P/S) were obtained from Gibco (Gaithersburg, MD, USA). Calf serum (CS) was supplied by HyClone (Logan, UT, USA). Earle’s balanced salt solution (EBSS), G418 disulfate salt, 3-MA, rapamycin, chloroquine diphosphate salt, diethylumbelliferyl phosphate (DEUP), 3-isobutyl-1-methylxanthine (IBMX), dexamethasone, insulin, ammonium chloride and leupeptin (NL), and Oil Red-O were offered by Sigma–Aldrich (St. Louis, MO, USA). Phos-tagTM, BCA protein assay kit, Lipofectamine 3000 Reagent and SuperSignal West Femto Maximum Sensitivity Substrate were purchased from Thermo Fisher Scientific (Grand Island, NY, USA). Plasmids of pEGFP-LC3 were obtained from Addgene (Cambridge, MA, USA). Plasmids of mRFP-GFP-LC3, RIPA lysis buffer were supplied by Beyotime (Shanghai, China). Triton X-100 and PVDF membranes were offered by Bio-Rad (Hercules, CA, USA).

### 4.2. 3T3-L1 Cell Culture and Differentiation

3T3-L1 preadipocytes were obtained from ATCC (Manassas, VA, USA) and maintained in DMEM containing 10% CS and 1% P/S. Cells were differentiated into adipocytes as reported previously [44]. Briefly, 2 days post-confluent 3T3-L1 preadipocytes were stimulated with DMEM supplemented with 10% FBS, 1 μM dexamethasone, 0.5 mM IBMX and 5 μg/mL insulin for 2 days. Cells were subsequently cultured in maintaining medium (DMEM supplemented with 10% FBS and 5 μg/mL insulin) for 6 days. The medium was changed every other day. The fully differentiated 3T3-L1 cells was checked by microscopic observation and Oil-Red O staining. The fully differentiated cells were pre-treated with 100 μM DEUP, 10 mM 3-MA, NL (20 mM ammonium chloride and 100 μM leupeptin), or 2 μM CC as indicated, or starved in EBSS for 16 h before harvest.

### 4.3. Generation of SIRT3 Overexpression and SIRT3 Silenced Cell Lines

The pcDNA3.1-SIRT3 plasmid was generated in our previous study [45]. Briefly, 3T3-L1 cells (4 × 10^5^) were seeded in 35 mm plates and cultured for 24 h. The cells were transfected with 10 μg plasmids (pcDNA3.1 or pcDNA3.1-SIRT3) using Lipofectamine 3000 reagent. 24 h after transfection, 800 μg/mL G418 was added to select positive cells for 2 weeks.

3T3-L1 cells at 50% confluence were transfected with 4 μg shRNA (shRNA targeting *SIRT3* (mouse, sc-61556) or scrambled shRNA (mouse, sc-108060, Santa Cruz Biotechnology, Santa Cruz, CA, USA) for 6 h according to the manufacturer’s protocol. After incubated with fresh medium for an additional 48 h, cells were selected with 10 μg/mL puromycin for two weeks. Thereafter, cells were pooled together for further experiments.

### 4.4. RNA Interference and Transient Infection

The siRNA targeting *Atg5*, *LAMP-2A*, *AMPKα1*, *AMPKα2*, *AMPKβ1*, and *AMPKβ2* (Appendix A) were obtained from GenePharma (Shanghai, China). Scrambled non-targeting siRNA was used as a negative control. 3T3-L1 cells (1 × 10^5^) were seeded at 6-well plates and cultured for 24 h. The cells were treated with 10 nM siRNA using Lipofectamine 3000. After 6 h, the cells were switched to fresh medium and incubated for an additional 24 h.

3T3-L1 cells (2 × 10^5^) were seeded on the coverslips in 6-well plates and cultured for 24 h. Then, the cells were infected with Ad-mCherry-GFP-LC3 (10 μL, multiplicity of infection = 5, #C3011, Beyotime, Shanghai, China) according to the manufacturer’s protocol. After 6 h, the cells were switched to fresh medium and incubated for an additional 24 h, following by fluorescence detection.

### 4.5. Immunoblotting

3T3-L1 adipocytes were lysed with RIPA lysis buffer supplying 1% protease inhibitor cocktail and 1% phenylmethane sulfonylfluoride (Sigma-Aldrich, St. Louis, MO, USA). BCA Protein Assay Kit was used to determine protein concentration. An equal amount of protein (15‒30 μg) was separated using 5%–12% SDS-PAGE and then transferred to PVDF membranes. After blocking with 5% nonfat milk for 2 h at room temperature, the membranes were probed with specific primary antibodies (Appendix A) overnight at 4 °C and then probed with corresponding secondary antibodies for 1 h at room temperature. Phos-tagTM SDS-PAGE was performed according to the manufacturer’s instructions. Signals were developed using a SuperSignal West Femto Maximum Sensitivity Substrate kit. Then, specific protein bands were visualized by the ChemiDoc MP Imaging System (Bio-Rad, Hercules, CA, USA) and quantitated with Image Lab 5.1 (Bio-Rad, Hercules, CA, USA) as described previously [46].

### 4.6. Nile Red Staining

Nile red staining was performed as described previously [47]. Briefly, 3T3-L1 cells were fixed with 10% formaldehyde solution and stained with Nile red (1 μg/mL) for 30 min at 4 °C. After washed with PBS thrice, intracellular Nile red-stained LDs were observed using fluorescence microscopy and quantitated using a flow cytometer with excitation and emission wavelength at 530 nm and 590 nm, respectively.

### 4.7. Isolation of LD Fractions

LD fractions from 3T3-L1 adipocytes was isolated as described in [8]. Cells were lysed in 0.25 M sucrose and centrifuged at 6800× *g* for 5 min at 4 °C. Supernatants including the fatty layer were transferred to a new tube and centrifuged at 17,000× *g* for 10 min at 4 °C to eliminate unwanted cellular debris. Subsequently, the supernatant was transferred to a new tube, adjusted to 20% sucrose and centrifuged in a discontinuous sucrose density gradient at 27,000× *g* for 30 min at 4 °C. LD fraction was delipidated using successive washes in acetone and ether, and solubilized in 2% SDS for immunoblotting [48].

### 4.8. Immunoprecipitation

Cell lysates (3 mg protein) were incubated with the indicated antibody (2 μg) at 4 °C overnight. Subsequently, 20 μL protein A/G-agarose beads (Santa Cruz Biotechnology, Santa Cruz, CA, USA) were added to the cell lysate and incubated on a rotator for 4 h at 4 °C. The beads were washed twice with PBS and then twice with lysis buffer supplemented with complete mini-protease inhibitor cocktail. Bound proteins were boiled in sample preparation buffer for 5 min.

### 4.9. Immunofluorescence staining

3T3-L1 adipocytes were grown on collagen-precoated glass coverslips and infected with a lentivirus expressing GFP-LC3 72 h before various treatments. Cells were fixed with 4% paraformaldehyde for 20 min, washed with PBS (pH 7.4) and then permeabilized with 0.5% Triton X-100 for 20 min at room temperature. The slides were then incubated with a primary antibody (1:100 dilution) at 4 °C overnight. The slides were washed with PBS and incubated with Texas Red-conjugated anti-rabbit IgG secondary antibody (1:1000 dilution) at room temperature for 1 h. The autophagic flux alterations and LDs morphology were detected using confocal microscopy (Olympus, Tokyo, Japan).

### 4.10. Determination of TG and T-CHO Levels

The levels of TG and T-CHO in 3T3-L1 cells were determined using commercial assay kits (Nanjing Jiancheng, Nanjing, Jiangsu, China) in accordance with the manufacturer’s protocols. The levels of TG and T-CHO were normalized by total protein amount.

### 4.11. Isoproterenol-Induced Lipolysis

The lipolysis activity of 3T3-L1 cells were measured as described previously [49]. Briefly, the cells were incubated with 10 μM isoproterenol (Sigma-Aldrich, St. Louis, MO, USA) as the stimulated condition, or DMSO as the basal condition for 2 h. Then, medium was collected and heated at 85 °C for 10 min. After briefly spinning down, 10 μL clear supernatant was used to measure the glycerol content using Free Glycerol Reagent (Sigma-Aldrich, St. Louis, MO, USA). The glycerol concentration was further normalized by protein concentration.

### 4.12. Seahorse Analysis

The OCR was determine by using a Seahorse Bioscience XF24-3 Extracellular Flux Analyzer (Agilent, Santa Clara, CA, USA), as described previously [26]. 3T3-L1 cells were seeded in XF24-well microplates (Seahorse Bioscience, Billerica, MA, USA) at 5 × 10^5^ cells per well. The fully differentiated cells were incubated in XF assay medium (low-buffered bicarbonate-free DMEM, pH 7.4) in the absence of CO_2_ for 1 h. After measuring basal OCR, 1 μM carbonyl cyanide-ptrifluoromethoxyphenylhydrazone (FCCP) and oligomycin were introduced in real time, respectively. After detection, cellular protein content was quantitated with a BCA kit. OCR were further normalized by the protein content.

### 4.13. High-Fat Diet (HFD)-Fed Mice

All the experimental protocols were in accordance with the National Institutes of Health guidelines for the Care of Use of Laboratory Animals, and approved by the Animal Ethical and Welfare Committee of University of Macau (No. ICMS-AEC-2014-06, 23 June 2014). Male C57BL/6J mice were purchased from Faculty of Health Sciences, University of Macau, and housed at 22 ± 1 °C under a 12 h light, 12 h dark cycle with 50% humidity, and fed with a regular chow diet (Guangdong Medical Lab Animal Center, Guangzhou, Guangdong, China) and water ad libitum under standard conditions (specific-pathogen-free). The mice (8 weeks old) were randomly separated into two groups, each of 6 mice. One group of mice (RD) were fed with a regular chow diet (calorie, 2.35 kcal/g), and the other group of mice were fed with a 45% HFD (calorie, 4.5 kcal/g, Trophic Animal Feed High-Tech Co, Nantong, Jiangsu, China) for 16 weeks. Epididymal WAT was collected after 16 h fasting and stored at −80 °C.

### 4.14. Statistical Analysis

Data were expressed as mean ± SEM based on at least three independent experiments and analyzed on Graphpad Prism 6.0 (GraphPad Software, San Diego, CA, USA). The significance of differences between groups were assessed by ANOVA test. *p* < 0.05 indicated the presence of a statistically significant difference.

## 5. Conclusions

In summary, our study found that SIRT3 promotes macroautophagy and CMA via activating AMPK, which stimulates LD degradation and attenuates lipid accumulation in adipocytes (Figure 7). It is the first report providing evidence that SIRT3 plays a positive role in macroautophagy and CMA to promote lipid mobilization in adipocytes. LDs can be efficiently degraded by CMA and then incorporated into autophagosomes, ultimately resulting in the release of FFAs as fuel for mitochondrial β-oxidation. Macroautophagy and CMA play an important role in the protective effect of SIRT3 against lipid accumulation in adipocytes, and SIRT3 might be a therapeutic target to optimize obesity treatment.

## Figures and Tables

**Figure 1 ijms-21-00372-f001:**
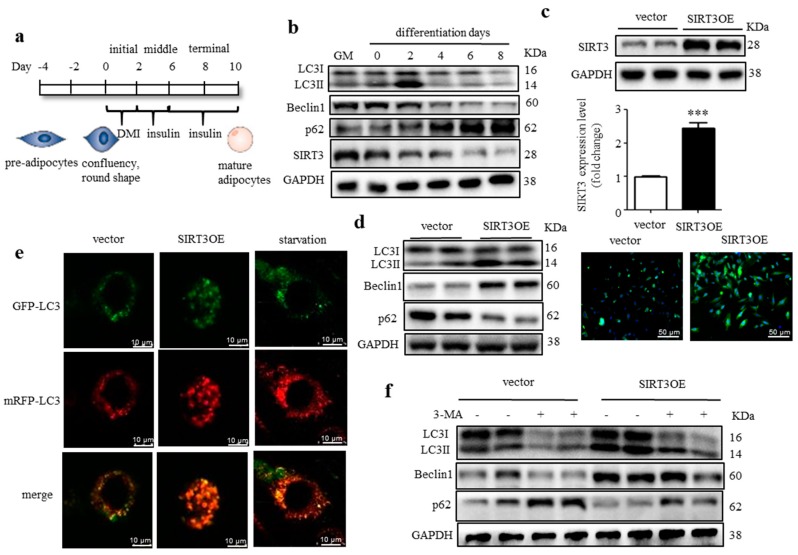
Macroautophagy is dynamically regulated during adipocyte differentiation and SIRT3 positively regulates macroautophagy in adipocytes. (**a**) The procedure of 3T3-L1 adipocyte differentiation; (**b**) Expression of SIRT3 and macroautophagy related proteins during adipocyte differentiation were analyzed by Western blots; (**c**) Generation of SIRT3 overexpressed 3T3-L1 cell line. SIRT3 protein level was detected by western blots and immunofluorescence (green), scale bar = 50 μm. (**d**) Expression of macroautophagy-related proteins in SIRT3OE and vector cells was analyzed by Western blots. GAPDH was used as a loading control; (**e**) SIRT3OE and vector 3T3-L1 cells were transiently infected with the mRFP-GFP-LC3 lentivirus. mRFP-GFP-LC3 puncta were examined using a confocal microscope. Scale bar = 10 μm. (**f**) SIRT3OE and vector cells were treated with 3-MA for 24 h. Expression of macroautophagy related proteins was analyzed by Western blots. GAPDH was used as a loading control. Data represented means ± S.D., *n* = 6, *** *p* < 0.001 SIRT3OE versus vector cells.

**Figure 2 ijms-21-00372-f002:**
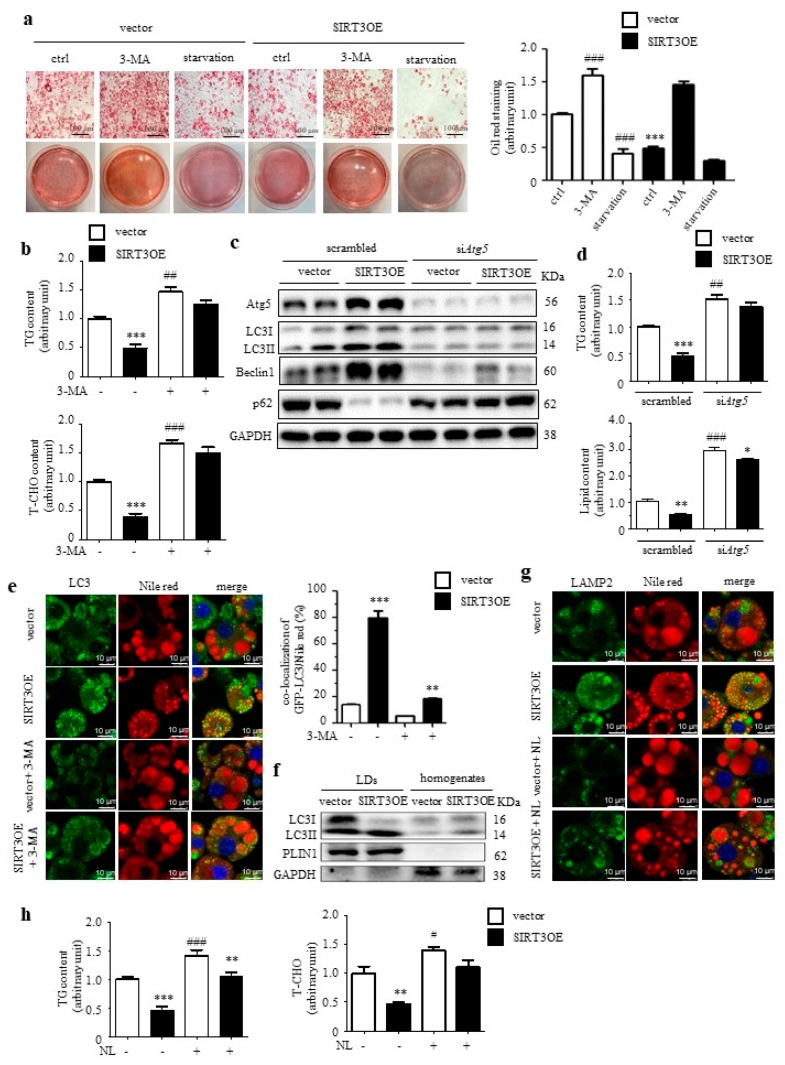
Overexpression of SIRT3 decreases lipid accumulation in adipocytes through activating macroautophagy. (**a**) The vector and SIRT3OE cells were treated with DMSO (ctrl), 3-MA, or Earle’s balanced salt solution (EBSS, starvation) for 16 h. Representative Oil Red O staining images and the relative intensity. Scale bar = 100 μm. (**b**) TG and T-CHO levels in vector and SIRT3OE cells treated with or without 3-MA for 16 h. (**c**) SIRT3OE and vector cells were transfected with si*Atg5*. Expression of macroautophagy-related proteins was analyzed by Western blots. GAPDH was used as a loading control. (**d**) TG level and lipid content in vector and SIRT3OE cells transfected with or without si*Atg5*. (**e**) Immunofluorescence of LC3 (green) and fluorescence of Nile Red (red) in SIRT3OE and vector cells treated with or without 3-MA. Nuclei were stained with DAPI (blue). Scale bar = 10 μm. The Mander’s overlapping coefficient of LC3 and Nile Red was calculated. (**f**) LC3 protein level in LD fraction and homogenates from SIRT3OE and vector cells. (**g**) Immunofluorescence of LAMP2 (green) and fluorescence of Nile Red (red) in SIRT3OE and vector cells treated with or without NL. Nuclei were stained with DAPI (blue). Scale bar = 10 μm. (**h**) TG and T-CHO levels in vector and SIRT3OE cells treated with or without NL for 16 h. Data represented means ± S.D., *n* = 6, * *p* < 0.05, ** *p* < 0.01 and *** *p* < 0.001 SIRT3OE versus vector cells. # *p* < 0.05, ## *p* < 0.01 and ### *p* < 0.001 vehicle control versus treated cells.

**Figure 3 ijms-21-00372-f003:**
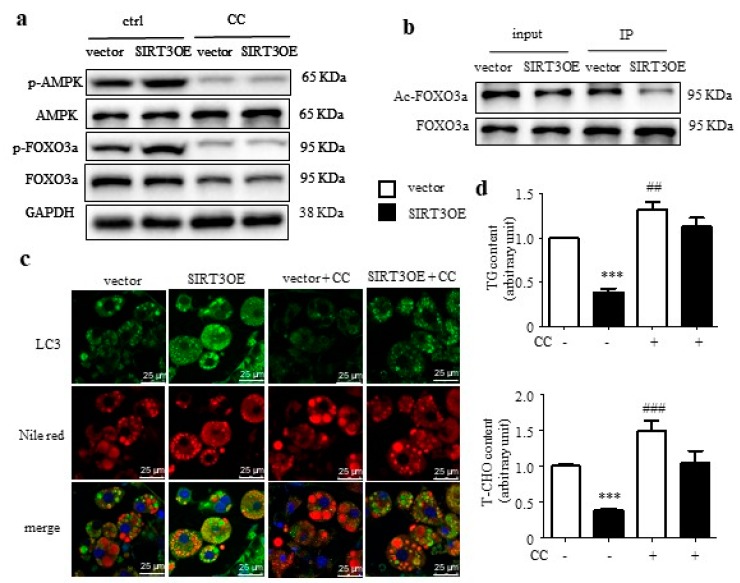
SIRT3-induced macroautophagy is mediated through activating AMPK in adipocytes. (**a**) p-AMPK, AMPK, p-FOXO3a, and FOXO3a expression in vector and SIRT3OE cells treated with or without CC. GAPDH was used as a loading control. (**b**) Acetylated and total FOXO3a protein levels in vector and SIRT3OE cells. (**c**) Immunofluorescence of LC3 (green) and fluorescence of Nile Red (red) in SIRT3OE and vector cells treated with or without CC. Nuclei were stained with DAPI (blue). Scale bar = 25 μm. (**d**) TG and T-CHO levels in vector and SIRT3OE cells treated with or without CC for 16 h. Data represented means ± S.D., *n* = 6, *** *p* < 0.001 vector versus SIRT3OE cells. ## *p* < 0.01 and ### *p* < 0.001 vehicle control versus CC treated cells.

**Figure 4 ijms-21-00372-f004:**
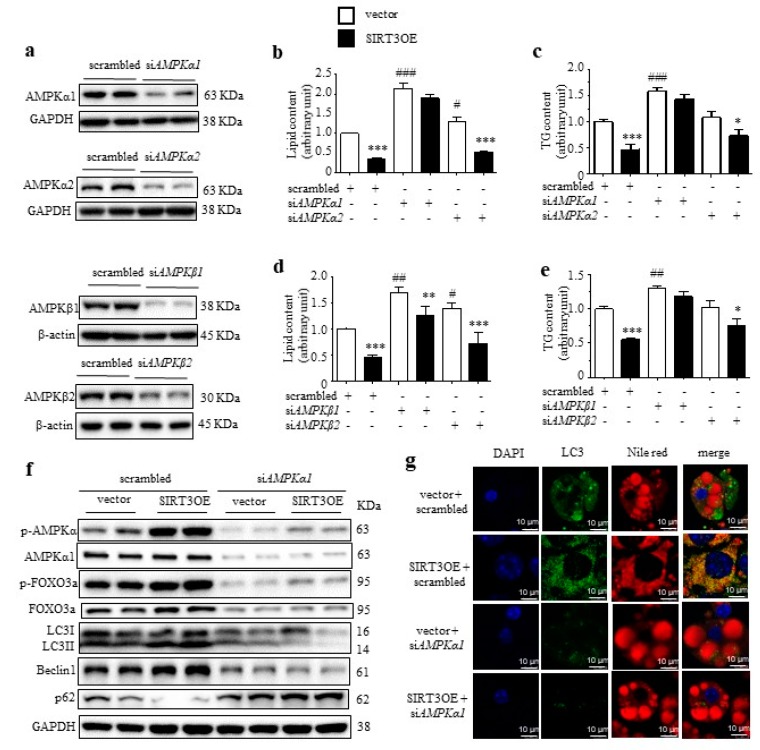
Knockdown of *AMPKα1* reversed SIRT3-induced activation of macroautophagy. (**a**) Generation of si*AMPKα1*, si*AMPKα2*, si*AMPKβ1* and si*AMPKβ2* cell lines. The lipid content (**b**) and the cellular TG content (**c**) in vector and SIRT3OE adipocytes with silenced *AMPKα1* or *AMPKα2*. * *p* < 0.05 and *** *p* < 0.001 vector versus SIRT3OE cells. # *p* < 0.05 and ### *p* < 0.001 scrambled vs. si*AMPKα1*/si*AMPKα2*. The lipid content (**d**) and the cellular TG content (**e**) in vector and SIRT3OE adipocytes with silenced *AMPKβ1* or *AMPKβ2*. * *p* < 0.05, ** *p* < 0.01, and *** *p* < 0.001 vector versus SIRT3OE cells. # *p* < 0.05 and ## *p* < 0.01 scrambled vs. si*AMPKβ1*/si*AMPKβ2*. (**f**) LC3, Beclin1, p62, phosphorylated AMPK, total AMPK, phosphorylated FOXO3a and total FOXO3a protein levels in vector and SIRT3OE adipocytes with or without silenced *AMPKα1*. GAPDH was used as a loading control. (**g**) Immunofluorescence of LC3 (green) and fluorescence of Nile Red (red) in SIRT3OE and vector cells with or without silenced *AMPKα1*. Nuclei were stained with DAPI (blue). Scale bar = 10 μm. Data represented means ± S.D., *n* = 6.

**Figure 5 ijms-21-00372-f005:**
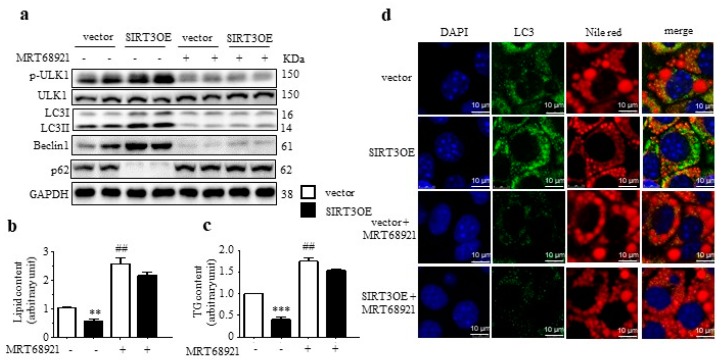
SIRT3 activates macroautophagy in adipocytes through the AMPK-ULK1/2 pathway. (**a**) LC3, Beclin1, p62, phosphorylated ULK1 and total ULK protein levels in vector and SIRT3OE adipocytes treated with or without MRT68921. GAPDH was used as a loading control. The lipid content (**b**) and the cellular TG content (**c**) in vector and SIRT3OE adipocytes treated with or without MRT68921. (**d**) Immunofluorescence of LC3 (green) and fluorescence of Nile Red (red) in vector and SIRT3OE adipocytes treated with or without MRT68921. Nuclei were stained with DAPI (blue). Scale bar = 10 µm. Data are shown as mean ± S.D., *n* = 6, ** *p* < 0.01, and *** *p* < 0.001 vector versus SIRT3OE cells. ## *p* < 0.01 vehicle control versus MRT68921 treated cells.

**Figure 6 ijms-21-00372-f006:**
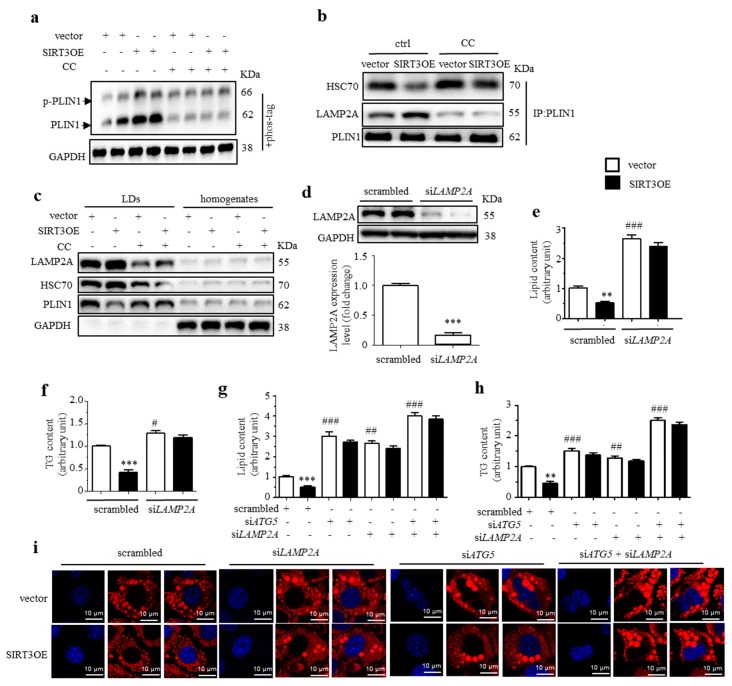
SIRT3 promotes CMA in adipocytes through activating AMPK. (**a**) The phosphorylated and total PLIN1 protein levels in SIRT3OE and vector cells treated with or without CC. GAPDH was used as a loading control. (**b**) The levels of co-precipitated HSC70 and LAMP2 with PLIN1 in vector and SIRT3OE cells treated with or without CC. (**c**) Expression of LAMP-2A, HSC70 and PLIN1 in LDs and homogenates from vector and SIRT3OE cells treated with or without CC. GAPDH was used as a loading control. (**d**) LAMP2A expression in the scrambled and si*LAMP2A* cell lines. *** *p* < 0.001 scrambled vs. si*LAMP2A*. The lipid content (**e**) and the cellular TG content (**f**) in vector and SIRT3OE adipocytes with or without silenced *LAMP-2A*. ** *p* < 0.01, and *** *p* < 0.001 vector versus SIRT3OE cells. # *p* < 0.05 and ### *p* < 0.001 scrambled vs. si*LAMP2A*. The lipid content (**g**) and the cellular TG content (**h**) in vector and SIRT3OE adipocytes with silenced *Atg5*, silenced *LAMP-2A* or simultaneously silenced *Atg5/LAMP-2A*. (**i**) Intracellular lipid in vector and SIRT3OE adipocytes with silenced *Atg5,* silenced *LAMP-2A* or simultaneously silenced *Atg5/LAMP-2A* was visualized with Nile red (red) staining. Nuclei were stained with DAPI (blue). Scale bar = 10 µm. ** *p* < 0.01, and *** *p* < 0.001 vector versus SIRT3OE cells. ## *p* < 0.01 and ### *p* < 0.001 scrambled vs. si*LAMP-2A*, si*Atg5* or si*LAMP-2A*/si*Atg5*. Data are shown as mean ± S.D., *n* = 6.

**Figure 7 ijms-21-00372-f007:**
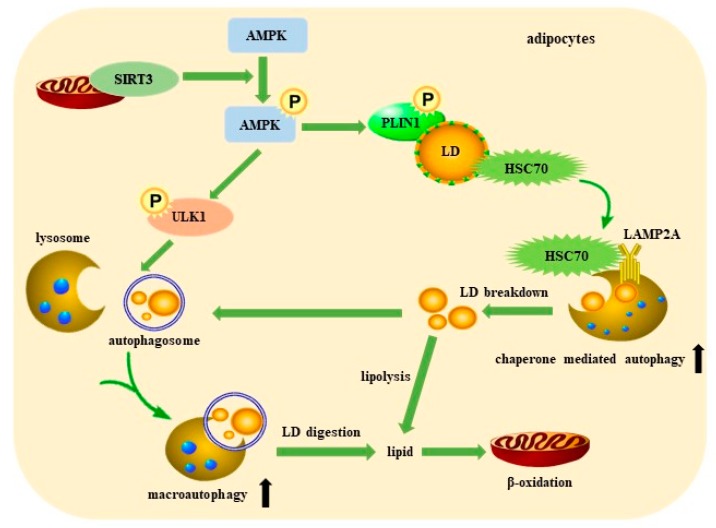
Schematic diagram of the role of SIRT3 in promoting lipid mobilization in adipocytes.

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
