# Peer review of "SIRT3 Acts as a Positive Autophagy Regulator to Promote Lipid Mobilization in Adipocytes via Activating AMPK"

_ijms, 2020, doi:10.3390/ijms21020372_

Round 1

Reviewer 1 Report

The main point of the present work is the proposal that SIRT3, via activating AMPK, acts as a positive regulator of macroautophagy and CMA in adipocytes. To show that, the authors use different strategies to inhibit autophagy which attenuate the effect of SIRT3OE on lipid content. In general, convincing evidences that SIRT3 plays a role in lipid mobilization in adipocytes are presented.  

Minor points

- An image showing the subcellular distribution of the exogenous SIRT3 should be included in the manuscript.

-In Fig.6, Nile Red images for siATG5 and siLAMP2A should be included to reinforce the data.

-Fig. 6a: misslead headlines. Lacks of a SIRT3OE condition without CC and the molecular weights might be wrong.

-Suppl. Fig. 3 lacks headlines.

-In Suppl. Fig 4 the authors confirm that 3-MA treatment block the effect of SIRT3 OE on lipid accumulation at the terminal stage of differentiation. However, the Oil Red staining images are not convincing.

Author Response

The main point of the present work is the proposal that SIRT3, via activating AMPK, acts as a positive regulator of macroautophagy and CMA in adipocytes. To show that, the authors use different strategies to inhibit autophagy which attenuate the effect of SIRT3OE on lipid content. In general, convincing evidences that SIRT3 plays a role in lipid mobilization in adipocytes are presented.  
A: Thanks for the reviewer’s positive comments.

1) An image showing the subcellular distribution of the exogenous SIRT3 should be included in the manuscript.

A: Thanks for reviewer’s critical reminder and valuable suggestion. Based on the reviewer’s
suggestion, the immunofluorescence staining of SIRT3 was performed in SIRT3OE and vector adipocytes. As expected, SIRT3 protein is mainly expressed in cytosol, but not nucleus. These results were added in the Figure 1c.

2) In Fig.6, Nile Red images for siATG5 and siLAMP2A should be included to reinforce the data.

A: Thanks for reviewer’s critical reminder. According to the reviewer’s suggestion, Nile red staining were performed, and the results supported that simultaneous silencing of Atg5 and LAMP-2A resulted in more severe lipid accumulation in adipocytes, compared with either Atg5 knockdown or LAMP-2A knockdown cells, and almost reversed the SIRT3’s effect on reduction of lipid accumulation in adipocytes. The Nile red staining images were added in the Figure 6i.

3) Fig. 6a: misslead headlines. Lacks of a SIRT3OE condition without CC and the molecular weights might be wrong.

A: Apologize for our mistakes. In the Figure 6a, the first four lanes should be without CC treatment, and the last four lanes should be with CC treatment. Additionally, the molecular weight of p-PLIN1 should be 66 KDa. The Figure 6a has been replaced.

4) Suppl. Fig. 3 lacks headlines.

A: Thanks for the reviewer’s reminder. The headlines of Suppl. Fig. 3 was added in the figure legend as ‘ (b) Oil red-O staining of vector and SIRT3OE adipocytes. Cells were treated with DMSO (ctrl), 3-MA or HBSS (starvation), respectively, for 24 h. Oil red-O staining was performed on day 10 of differentiation’.

5) In Suppl. Fig 4 the authors confirm that 3-MA treatment block the effect of SIRT3 OE on lipid accumulation at the terminal stage of differentiation. However, the Oil Red staining images are not convincing.

A: Thanks for the reviewer’s reminder. We re-did the Oil Red-O staining and replaced the images in the revised manuscript.

Reviewer 2 Report

In the given study, the authors have shown SIRT3 as a positive regulator of macroautophagy and CMA in 3T3-L1 adipocytes. Overexpression of SIRT3 activated macroautophagy, mainly on lipid droplets, by activating AAMPK/ULK1 pathway, which in turn resulting in reduced lipid accumulation. Moreover, SIRT3 overexpression induced the formation of PLN1/HSC70/LAMP2 complex, that activated CMA and caused the instability of lipid droplets in adipocytes. Overall, the findings are novel and it is well designed, and an elaborate study. Proper controls have been included, and most of the data supports the conclusions. The manuscript is acceptable in the present form.

Author Response

In the given study, the authors have shown SIRT3 as a positive regulator of macroautophagy and CMA in 3T3-L1 adipocytes. Overexpression of SIRT3 activated macroautophagy, mainly on lipid droplets, by activating AAMPK/ULK1 pathway, which in turn resulting in reduced lipid accumulation. Moreover, SIRT3 overexpression induced the formation of PLN1/HSC70/LAMP2 complex, that activated CMA and caused the instability of lipid droplets in adipocytes. Overall, the findings are novel and it is well designed, and an elaborate study. Proper controls have been included, and most of the data supports the conclusions. The manuscript is acceptable in the present form.

A: Thanks for the reviewer’s positive comments.